# Role of Muscarinic Acetylcholine Receptors in Intestinal Epithelial Homeostasis: Insights for the Treatment of Inflammatory Bowel Disease

**DOI:** 10.3390/ijms24076508

**Published:** 2023-03-30

**Authors:** Junsuke Uwada, Hitomi Nakazawa, Ikunobu Muramatsu, Takayoshi Masuoka, Takashi Yazawa

**Affiliations:** 1Department of Pharmacology, School of Medicine, Kanazawa Medical University, Uchinada 920-0293, Japan; 2Division of Genomic Science and Microbiology, School of Medicine, University of Fukui, Eiheiji 910-1193, Japan; 3Kimura Hospital, Awara, Fukui 919-0634, Japan; 4Department of Biochemistry, Asahikawa Medical University, Asahikawa 078-8510, Japan

**Keywords:** inflammatory bowel disease, epithelial barrier, homeostasis, acetylcholine, muscarinic receptor, non-neuronal acetylcholine

## Abstract

Inflammatory bowel disease (IBD), which includes Crohn’s disease and ulcerative colitis, is an intestinal disorder that causes prolonged inflammation of the gastrointestinal tract. Currently, the etiology of IBD is not fully understood and treatments are insufficient to completely cure the disease. In addition to absorbing essential nutrients, intestinal epithelial cells prevent the entry of foreign antigens (micro-organisms and undigested food) through mucus secretion and epithelial barrier formation. Disruption of the intestinal epithelial homeostasis exacerbates inflammation. Thus, the maintenance and reinforcement of epithelial function may have therapeutic benefits in the treatment of IBD. Muscarinic acetylcholine receptors (mAChRs) are G protein-coupled receptors for acetylcholine that are expressed in intestinal epithelial cells. Recent studies have revealed the role of mAChRs in the maintenance of intestinal epithelial homeostasis. The importance of non-neuronal acetylcholine in mAChR activation in epithelial cells has also been recognized. This review aimed to summarize recent advances in research on mAChRs for intestinal epithelial homeostasis and the involvement of non-neuronal acetylcholine systems, and highlight their potential as targets for IBD therapy.

## 1. Introduction

The intestinal lumen is exposed to various substances via food intake and other sources. Furthermore, a wide variety of bacteria reside in the intestinal lumen and form the gut microbiota. The intestinal epithelial cells separate the inner lamina propria from the luminal environment. Intestinal epithelial cells not only absorb essential nutrients but also form a barrier that blocks the invasion of unwanted antigens and bacteria. Disturbances in homeostasis and barrier function of the intestinal epithelium can result in the entry of bacteria and other antigens, thereby triggering an inflammatory response. Patients with inflammatory bowel disease (IBD), including Crohn’s disease and ulcerative colitis, often have impaired intestinal barrier function and exacerbated inflammation [1,2]. Current treatments for IBD generally aim to suppress inflammatory responses. In contrast, the maintenance or regeneration of the epithelial barrier results in removal of the cause of inflammation, which could be a viable strategy for the treatment of IBD. Therefore, the mechanisms that maintain the homeostasis of intestinal epithelial function should be clarified.

Gut function is governed by the autonomic nervous system. The parasympathetic nervous system, through its neurotransmitter, acetylcholine (ACh), activates muscarinic acetylcholine receptors (mAChRs) in intestinal smooth muscle cells to promote intestinal motility [3]. According to previous reports, mAChRs also exist in intestinal epithelial cells and regulate chloride (Cl^-^) secretion, which is important for mucosal hydration [4,5]. ACh also acts on nicotinic acetylcholine receptors (nAChRs). nAChRs exert anti-inflammatory effects in the gut. For example, stimulation of the vagus nerve activates α7-nAChRs expressed on macrophages in the spleen and gut, which suppress the release of pro-inflammatory cytokines, such as tumor necrosis factor-α (TNF-α) [6]. Therefore, nicotinic ACh signaling is a potential therapeutic target for IBD. On the other hand, in recent years, numerous reports have been published on the role of mAChRs in the maintenance of intestinal epithelial homeostasis. However, some of these reports contradict the function of mAChRs and the receptor subtypes involved. This review aimed to summarize the muscarinic receptor subtypes expressed in the intestinal epithelium and the role of mAChRs in intestinal epithelial homeostasis, with a focus on maintenance of the mucus layer, protection and regeneration of barrier function, and differentiation and proliferation of stem cells. In addition to the parasympathetic nerves, several other cell types secrete ACh to activate mAChRs in epithelial cells. As the mode of ACh synthesis and secretion by epithelial cells has been elucidated in recent years, reports on non-neuronal ACh systems were also reviewed. Finally, we discussed the potential of mAChRs as pharmacological targets for the treatment of IBD by protecting and improving intestinal epithelial function.

## 2. mAChRs in the Intestinal Epithelium

Muscarinic acetylcholine receptors (mAChRs) belong to the G protein-coupled receptor (GPCR) family and receive ACh as an endogenous ligand. In mammals, mAChRs consist of five subtypes (M_1–_M_5_). The M_1_, M_3_, and M_5_ subtypes couple to Gα_q/11_ proteins, which can activate phospholipase Cβ, leading to the production of diacylglycerol and inositol 1,4,5-trisphosphate, which activates protein kinase C (PKC) and mobilizes intracellular Ca^2+^, respectively. In contrast, the M_2_ and M_4_ receptors signal via Gα_i/o_ proteins to inhibit adenylyl cyclase activity and downregulate cyclic AMP production. mAChRs have been actively studied for a long time as they play a role in the regulation of motility in intestinal smooth muscle cells and the secretion of anions, such as chloride, in intestinal epithelial cells. The intestinal epithelium comprises aligned enterocytes/colonocytes, goblet cells, and Paneth cells. These cells differentiate from a common origin, the few leucine-rich repeat-containing G protein-coupled receptor 5 (Lgr5)-positive stem cells in the crypt base.

The expression of mAChR subtypes in these cells has been examined using immunohistochemistry, in situ hybridization, quantitative PCR, and pharmacological assays. In a previous review, Hirota et al. summarized the mAChR subtypes in the gut [7]. Here, we provide an update based on a recent study that evaluated mAChR subtypes with a focus on the intestinal epithelium (Table 1).

The mAChR subtypes that are mainly expressed in colonocytes are M_1_ and M_3_. In intestinal epithelial cells, mAChRs, especially the M_3_ subtype, act to promote Cl^-^ secretion [7]. Muscarinic toxin 7 (MT7) is a peptide antagonist derived from snake venom that is highly specific to the M_1_ subtype [21]. Experiments on the pharmacological binding of MT7 to colonic crypts have revealed that it consists of approximately 80% M_1_ and 20% M_3_ receptors. Functionally, M_3_ receptors promote Cl^-^ secretion, whereas M_1_ receptors may act in an inhibitory manner [14]. The human colonocyte cell-derived cell line, T84, which is often used as a model for intestinal epithelial cells, expresses both M_1_ and M_3_ [20], although the M_3_ subtype is responsible for Cl^-^ secretion [22]. However, as mAChR-stimulated ion secretion is maintained in M_3_ receptor-knockout (KO) mice, other subtypes, such as M_1_ receptors, may play compensatory roles [15,23]. Thus, both M_1_ and M_3_, expressed in colonocytes, couple to Gα_q/11_ proteins; however, their functions may be partially different.

Other epithelial cell types, such as Paneth cells, goblet cells, and stem cells, express mAChRs. Estimating the major mAChR subtype that is functional on the cell surface is a difficult task in pharmacological binding experiments owing to the relatively small population of these cells in the tissue. Therefore, studies evaluating the functional impact of cell-specific receptor KO mice or subtype-selective ligands are particularly important. In addition to the epithelial cells mentioned here, immune cells in the submucosal region express mAChRs, suggesting that ACh stimulation may indirectly affect epithelial cells via mAChRs on these immune cells. The relationship between mAChR subtypes expressed in these cells and their functions in epithelial homeostasis are described below.

## 3. Role of mAChRs in Mucus Layer Maintenance

Mucus and antimicrobial peptides (AMPs) on the surface of the epithelium prevent the invasion of macromolecules and bacteria before they reach the intestinal epithelium. Defects in the mucosal layer would facilitate bacterial infections and trigger inflammatory responses. Abnormalities in the mucosal barrier are frequently observed in patients with IBD [24]. Thus, improvement in mucus secretion and AMPs might be beneficial for IBD treatment.

Goblet cells, which reside in the intestinal crypts and villi, secrete mucus. ACh from the enteric nerve acts on mAChRs in goblet cells to induce mucus release [25,26]. The mAChR subtypes that are critical for the maintenance of the mucus layer have not been determined. In other tissues, the M_3_ subtype primarily mediates secretion in response to cholinergic stimulation. For example, in the conjunctival goblet cells, genetic disruption of the M_3_ subtype has been demonstrated to significantly reduce tear secretion [27]. Recent studies have suggested that the M_1_ subtype is primarily responsible for mucus secretion in the gut [28]. The expression of M_1_ in intestinal goblet cells was supported by a single-cell RNA-seq study [29]. The secretion of mucus from goblet cells is classified into compound exocytosis, in which multiple vesicles fuse in an intracellular Ca^2+^-dependent manner, and primary exocytosis, in which vesicles fuse individually. Mucus release by ACh stimulation is the former mechanism, and the process might be triggered by intracellular Ca^2+^ elevation by Gα_q/11_-coupled mAChR subtypes, such as M_1_ and M_3_ [30,31]. Furthermore, a study of M_3_ subtype KO mice revealed that M_3_ is important for the maintenance of goblet cells and the expression of Muc2, the main factor of mucus [32]. In contrast, Knoop et al. showed that M_4_ receptors are abundantly expressed with M_3_ receptors in goblet cells [12]. This research revealed that the M_4_ subtype plays an important role in immune tolerance in the small intestine through a mechanism called goblet cell-associated antigen passages (GAPs), which deliver luminal antigens to antigen-presenting cells in the lamina propria. In contrast, in the colonic mucosa, M_3_ receptors are involved in the induction of GAPs [28]. Thus, mAChR in goblet cells regulates the prevention of antigen entry by mucus secretion and the uptake and transport of antigens by GAPs.

Antimicrobial peptides are secreted from Paneth cells in the crypts of the small intestine. As AMPs prevent bacterial infections and control intestinal bacterial composites, their abnormalities are associated with pathologies, such as IBD [33]. Paneth cell secretion is affected by cholinergic stimuli as well as the bacterial milieu [34,35]. Stimulation of mAChRs in Paneth cells increases intracellular Ca^2+^ concentrations, which may induce the exocytosis of AMP-containing vesicles [36,37]. In *Caenorhabditis elegans*, cholinergic neurons induce Wnt expression in the intestinal epithelium via mAChR, leading to the upregulation of AMPs, such as C-type lectins and lysozymes [38]. Therefore, mAChRs may not only act on the secretion, but also on gene expression of AMPs.

Overall, mAChRs contribute to intestinal epithelial homeostasis by preventing the entry of macromolecules and bacteria into the epithelium via the release of mucus and AMPs.

## 4. Role of mAChRs in the Epithelial Barrier against Inflammatory Cytokines

Immune cells in the lamina propria are activated in response to bacterial and antigenic penetration. In the intestinal mucosa of IBD patients, the secretion of pro-inflammatory cytokines, such as TNF-α or interferon γ (IFNγ), is elevated [39,40,41]. These cytokines are known to increase the membrane permeability of intestinal epithelial cells and disrupt barrier functions [42,43]. Reducing the effects of these pro-inflammatory cytokines on the intestinal epithelium would be beneficial, as revealed by the efficacy of TNF-α-neutralizing antibodies in the treatment of IBD [44].

In rat colonic epithelium, the TNF-α/IFNγ-induced increase in paracellular permeability was prevented by prestimulation with a muscarinic agonist [19]. Nuclear factor κB (NF-κB) is a main signaling mediator of TNF-α and is involved in the dysregulation of intestinal epithelial integrity [45,46,47]. In HT-29/B6 cells, a human colorectal adenocarcinoma-derived cell line that can be used to study intestinal epithelial barrier function [48], TNF-α decreased barrier function and increased the expression of inflammatory cytokine (IL-8) in association with an increase in NF-κB signaling, both of which were suppressed by mAChR stimulation [19]. This attenuation of TNF-α effects is due to the shedding of the TNF-α receptors (TNFRs). Ectodomain shedding of TNFRs leads to the downregulation of TNF-α effects by reducing the density of cell surface TNFRs and neutralizing TNF-α with the cleaved, soluble form of TNFRs (sTNFRs) [49,50]. As HT-29/B6 cells predominantly express the M_3_ subtype, the action of TNF-α might be suppressed via this mAChR subtype. Tumor necrosis factor-α converting enzyme (TACE/ADAM-17) is a metalloprotease that can act on the shedding of TNFRs [51]. One upstream signaling molecule involved in TACE activation is p38 MAPK [52]. Based on further studies, the M_3_ subtype induces the reduction of TNF-α action by activating p38 MAPK following Gα_q/11_ protein-mediated Ca^2+^ responses, particularly store-operated calcium entry (SOCE) [53,54]. Activation of TACE by the M_3_ subtype also induces epidermal growth factor receptor (EGFR) transactivation [54], which may contribute to the optimal regulation of ion secretion [55] and intestinal epithelial homeostasis [56].

Stimulation of mAChRs confers resistance to interleukin-1β (IL-1β)-induced barrier dysfunction [57]. Treatment with IL-1β enhances the expression of chemokines (CXCL-1, CXCL-10, IL-8, and CCL-7) and myosin light-chain kinase (MLCK) through NF-κB signaling. Stimulation of mAChRs was not found to suppress the activation of NF-κB by IL-1β; therefore, the expression of these cytokines or MLCK was not suppressed. In contrast, mAChR stimulation markedly inhibited MLCK-mediated phosphorylation of the myosin light chain (MLC) by IL-1β; however, the mechanism is unclear. Activation of MLCK leads to barrier disruption through endocytosis of tight junction factors, including occludin [58,59]. In fact, IL-1β reduced the amount of occludin localized to the tight junction, which was inhibited by mAChR stimulation. Thus, mAChR may inhibit IL-1β-induced impairment of barrier function via the suppression of MLC phosphorylation by MLCK [57].

Dextran sulfate sodium (DSS)-induced colitis is widely used as a model of human ulcerative colitis [60]. The administration of DSS damages intestinal epithelial cells, resulting in the infiltration of macrophages and the release of pro-inflammatory cytokines, such as TNF-α and IL-1β, in response to invading bacteria. Neutralizing antibodies against TNF-α were found to improve mucosal integrity in the DSS-induced colitis model [61]. Notably, DSS-induced colitis was more severe in M_3_ KO mice [23]. Recently, McN-A-343 was reported to exhibit anti-inflammatory effects in acetic acid-treated mice, another experimental model of ulcerative colitis [62]. McN-A-343 is widely used as an M1 selective agonist owing to its relatively high efficacy against the M1 subtype; however, it also acts on other mAChR subtypes [63]. Therefore, mAChRs, especially the M_1_ and M_3_ subtypes, may reduce the action of pro-inflammatory cytokines, such as TNF-α and IL-1β, released by the activated innate immune system and prevent the disruption of the intestinal epithelial barrier.

Hosic and colleagues established a primary culture system of human small intestinal epithelial cells, rather than immortalized cell lines, to determine the effects of TNF-α on barrier function [64]. In their study, neither nicotinic nor muscarinic stimulation improved TNF-α-induced barrier disruption. Therefore, it may be necessary to reconsider the role of mAChRs in epithelial cells by comparing the expressed mAChR subtypes and responsiveness between this primary culture system, cell lines, and in vivo.

## 5. Role of mAChRs in Epithelial Barrier Repair

As disruption of epithelial barrier function allows continued macromolecular and bacterial invasion, repair of the barrier function is important to prevent further inflammatory responses. In a porcine colonocyte culture system, stimulation with carbachol or oxotremorine increased transepithelial resistance during the establishment of epithelial barrier function [65]. This effect was inhibited by co-treatment with atropine, suggesting that muscarinic stimulation facilitates epithelial barrier formation, and mAChRs may play a role in the regeneration of the damaged epithelial barrier. Colonocyte-derived ACh was also detected, and the administration of atropine without muscarinic agonists suppressed the establishment of barrier function. Therefore, the autocrine/paracrine action of colonocyte-derived non-neuronal ACh may be involved in barrier formation [65].

Studies using T84 cells derived from human colonocytes expressing the M_1_ and M_3_ subtypes have reported that M_1_ receptor-mediated extracellular signal-regulated kinase 1/2 (ERK1/2) activation and focal adhesion kinase (FAK) activation contribute to recovery from ethanol-induced epithelial injury [20]. FAK activity is important for epithelial barrier maintenance and repair through the regulation of tight junction complex redistribution [66]. Activation of ERK1/2 also facilitates intestinal barrier function via expression of the tight junction factors [67,68]. The importance of the M_1_ subtype, rather than M_3_, for ERK1/2 activation has been demonstrated using rodent colonic mucosal fragments [14]. Khan et al. showed that the expression of the M_1_ subtype was reduced by IFNγ-induced barrier perturbation in T84 cells [20]. This observation suggests that inflammatory cytokines inhibit barrier restoration by suppressing M_1_ receptor levels. Tyrosine kinase inhibitors used as anticancer drugs have been demonstrated to reduce the barrier function of T84 cell monolayers, whereas cholinergic stimulation with carbachol delays barrier function impairment through ERK1/2 activation [69]. Thus, signaling through ERK1/2 and FAK may contribute to mAChR-induced epithelial barrier maintenance or repair. These kinases are involved in cell proliferation and migration as well. Therefore, the activation of these kinases by mAChRs is expected to participate in the replenishment of epithelial cells from stem and progenitor cells, and the maintenance of epithelial cell homeostasis, as described below.

Although mAChRs are important for maintaining barrier function against injury or restoring impaired barrier function, they may not be essential for constitutive epithelial barrier function, at least in the colon. This notion is because genetic ablation of M_1_, M_3_ or M_1_/M_3_ showed no significant difference in intestinal epithelial permeability [32,70]. However, in the small intestine, the constitutive barrier function is partially impaired in M_3_ KO mice [70]. Therefore, the role of mAChRs in barrier function may differ in the small and large intestine.

## 6. Role of mAChRs in Epithelial Cell Regeneration

The differentiation and proliferation of stem and progenitor cells are essential for the maintenance of intestinal epithelial cells, which undergo rapid turnover and replenishment of epithelial cells after injury. Released ACh from enteric cholinergic neurons upregulates mucosal growth in a scopolamine-sensitive manner, suggesting that mAChRs may facilitate these processes [16,71]. The stem cells responsible for intestinal epithelium regeneration are Lgr5-positive cells located at the bottom of the crypts. Several studies have revealed the mAChR subtypes expressed in Lgr5-positive stem cells. Greig et al. detected only the M_1_ subtype in jejunal and ileal crypt-based samples using RT-PCR [13]. In contrast, immunohistochemical studies have suggested the expression of the M_3_ and M_5_ subtypes [10,16]. Importantly, scopolamine treatment inhibited the differentiation and proliferation of Lgr5-positive cells, and caused their disappearance from the crypt base of the small intestine [10]. In addition, administration of the M_1_-selective agonist McN-A-343 promoted cell proliferation and increased intestinal mucosal growth in mice [13]. Therefore, mAChRs may contribute to the maintenance of stem cell function in the intestinal epithelium.

Conversely, some reports have revealed the negative effects of mAChRs on intestinal stem cell function. Treatment with the mAChR agonist inhibited organoid proliferation and differentiation in crypt-villus organoid cultures of the small intestine [8,9], while treatment with atropine alone promoted them. Such findings indicate that mAChRs inhibit the differentiation and proliferation of stem and progenitor cells by receiving non-neuronal ACh secreted by the intestinal epithelial cells themselves [9]. In addition, M_3_ KO mice displayed an increase in crypt size and the facilitation of cell proliferation and migration [72]. The increased villus height has been observed in conventional mAChR knockout mice, especially in M_2_, M_3_, and M_5_ receptor-KO mice [73].

Taken together, stimulation with M_3_ and M_5_ receptors may have inhibitory effects on differentiation and proliferation, whereas the M_1_ receptors contribute to epithelial cell regeneration and maintenance by maintaining and promoting the function of Lgr5-positive stem and progenitor cells. This proposal is intriguing as the M_1_, M_3_, and M_5_ subtypes are coupled to the same Gα_q/11_ protein. One possibility is that one of the M_1_ or M_3_/M_5_ subtypes is expressed not on Lgr5-positive stem cells, but on cells that form a stem cell niche near the stem cells, and indirectly regulates stem cell function. Alternatively, this phenomenon may be related to prior findings that the activities of ERK1/2 and FAK, which contribute to cell proliferation and migration in intestinal epithelial cells, are selectively induced by the M_1_ receptor rather than by the M_3_ receptor [14,20]. In fact, ERK1/2 inhibitors have been reported to suppress cell proliferation by inhibiting enhanced ERK1/2 activity in the intestinal organoids of M_3_ KO mice [72]. It is unclear why such a difference exists between M_1_- and M_3_-mediated signaling. In experiments performed with varying amounts of the Gα_q/11_ protein in the intestinal epithelial cell line, Gα_q/11_ signaling was suggested to act in a growth-suppressive manner and may be responsible for physiological effects, such as secretion and absorption in terminally differentiated intestinal epithelial cells [74]. Thus, the M_1_ receptor may activate ERK1/2 in a Gα_q/11_ protein-independent pathway. On the other hand, deletion of Gα_q/11_ in intestinal epithelial cells impairs proper differentiation, particularly into Paneth cells, suggesting that the effects of M_1_/M_3_-mediated Gα_q/11_ signaling on differentiation regulation require further studies [75].

Overall, an mAChR subtype-specific balance regulation may exist between the promotion and inhibition of proliferation and differentiation for intestinal epithelial regeneration.

## 7. Involvement of mAChRs in Epithelial Barrier Impairment

As described above, mAChRs in the intestinal epithelial cells contribute to epithelial homeostasis through various mechanisms. However, several studies have reported that mAChRs may reduce epithelial barrier function. For example, experiments on rat ileal segments have revealed that carbachol treatment enhances transport from the mucosa to the plasma membrane via endocytosis and the paracellular pathway [76]. Furthermore, electrical stimulation of the vagus nerve led to permeability of the jejunal epithelium through the activation of mAChRs [26]. Notably, the passage of bacteria increased when the small intestinal epithelium, stimulated with mAChRs via the intraperitoneal administration of pilocarpine, was examined in the Ussing chamber [77]. This study also revealed a pilocarpine-induced decrease in mucus secretion. In the mouse ileum, the M_3_ subtype increased the barrier permeability of macromolecules [78]. In contrast, mAChR activation in primary cultured intestinal epithelial cells or colonic cell lines, such as Caco-2 and T84 cells, was not found to exacerbate epithelial barrier function [57,64,65,79].

The presence of various cell types in the intestinal tissue could explain this difference. In addition to the epithelium, the tissue used in the Ussing chamber also contains immune and nervous system cells in the submucosal layer. Stress has been demonstrated to increase intestinal epithelial permeability via the activation of mAChRs by cholinergic neurons. Corticotropin-releasing factor (CRF) and the activation of mast cells play important roles in this process [80,81]. Wallon et al. demonstrated the presence of eosinophils expressing muscarinic M_2_ and M_3_ receptors in the subepithelial regions. The release of CRF by eosinophils via the activation of M_3_ receptors reduces intestinal epithelial barrier function by activating neighboring mast cells [79]. This research also found increased levels of activated and degranulated eosinophils in patients with ulcerative colitis, suggesting a link between mAChR stimulation-mediated eosinophil activation and the exacerbation of inflammation. In addition, activation of M_3_ receptors on macrophages leads to their differentiation into classically activated phenotypes, thereby facilitating inflammatory responses [32].

These studies indicate that the activation of mAChRs in a tissue-wide or subtype-unselective manner may have deleterious effects on intestinal epithelial barrier function. Therefore, local or subtype-specific activation of mAChRs is important for epithelial homeostasis.

## 8. Non-Neuronal Acetylcholine

Multiple sources of ACh—including ACh released from parasympathetic or enteric cholinergic nerves, and non-neuronal ACh released from intestinal epithelial cells and T cells—act on intestinal epithelial cells and their surrounding cells. Acetylcholine from parasympathetic and enteric neurons act on macrophage α7 nicotinic receptors to suppress pro-inflammatory cytokine release and stimulate mucus secretion from goblet cells via mAChR activation [26,82]. Neuronal ACh may also act on cryptic basal stem and progenitor cells, promoting intestinal mucosal tissue growth [16,71].

The importance of ACh released from non-neuronal cells in several tissues has been gradually recognized [83]. As ACh is an unstable substance that is easily degraded by acetylcholinesterase (AChE), ACh from nerve endings may be insufficient to activate receptors on epithelial cells. It has therefore been suggested that ACh released by epithelial cells may act in an autocrine/paracrine manner. Indeed, non-neuronal ACh has been demonstrated to be involved in epithelial barrier formation and the regulation of mucosal growth [9,65].

Choline acetyltransferase (ChAT) is responsible for ACh synthesis from choline. Transgenic mice expressing fluorescent proteins under the control of the ChAT promoter showed that ChAT-positive cells exist as scattered solitary cells in the epithelium of the small intestine and colon [84]. These tuft cells expressing ChAT were not found to express the vesicular ACh transporter (vAChT), except in the proximal colon, suggesting that ACh was released in a vesicle-independent manner. Consistently, vesamicol, an inhibitor of vAChT, was not found to inhibit non-neuronal ACh secretion in the colonic epithelium [85]. The polyspecific organic cation transporter OCTN1 has a variant (amino acid substitution L503F) associated with Crohn’s disease [86]. Interestingly, OCTN1 can transport ACh, and its transport function has been demonstrated to be reduced in the L503F variant [87]. Therefore, OCTN1 is one of the possible candidates responsible for the release of non-neuronal ACh in intestinal epithelial cells. As tuft cells do not express a high-affinity choline transporter (CHT1), they may provide choline for ACh synthesis via a different mechanism from cholinergic neurons [84]. The choline transporter-like (CTL) family consists of five members, of which CTL4 has been reported to contribute to choline uptake associated with ACh synthesis and secretion [88]. The expression of both CTL4 and ChAT increased when an inflammatory response was induced in the mouse ileum by lipopolysaccharide (LPS) treatment [89]. Therefore, inflammation may increase non-neuronal ACh production in the intestinal epithelium. However, in a previous study, ChAT expression was decreased in the intestinal epithelium in specimens from patients with Crohn’s disease and ulcerative colitis [57].

CD4+ T cells that infiltrate the gut express ChAT and synthesize ACh. ChAT-positive T cells participate in host defense against bacterial infections [90]. In addition, T cell-derived ACh has been reported to contribute to the expression of AMPs, including lysozymes and defensin A [91]. In DSS colitis models, ChAT-positive T cells exacerbate the acute immune response but support the later resolution of intestinal inflammation [92].

As described above, neuronal and non-neuronal ACh systems differ in certain factors involved in the choline uptake and ACh release mechanisms. Therefore, neuronal and non-neuronal ACh levels may be pharmacologically distinguished and regulated by targeting these differences.

## 9. mAChRs as a Potential Therapeutic Target for IBD

As summarized in Figure 1, the activation of mAChRs in intestinal epithelial cells is effective in the maintenance of homeostasis. However, as each mAChR subtype is expressed in different cell types and plays different roles, the outcome may vary depending on the subtype activated. For example, M_1_ receptors have been demonstrated to be responsible for epithelial barrier regeneration and mucus secretion, while M_3_ receptors are involved in the protection against inflammatory cytokines. The activation of M_3_ receptors expressed on eosinophils and macrophages may indirectly threaten epithelial barrier function. In addition, the migration and invasion of colon cancer cells are stimulated by the activation of M_3_ receptors [93], whose expression is increased in cancer cells [94]. In this context, the activation of M_3_ receptors may adversely affect intestinal epithelial homeostasis. Therefore, it is important to determine the effects of subtype-selective agonists and positive allosteric modulators (PAMs) on intestinal epithelial cell homeostasis and their impact on IBD treatment. In particular, M_1_ receptor-selective agonists and PAMs are being actively developed as potential therapeutic agents for schizophrenia and Alzheimer’s disease [95,96]. Exploring the potential of these drugs in the treatment of IBD is an interesting challenge. Gastrointestinal disorders, such as diarrhea, are major adverse effects of muscarinic stimulation. This effect is mainly due to the M_3_ subtype in smooth muscle, despite reports of the involvement of the M_1_ and M_2_ subtypes [97,98]. Therefore, the stimulation of epithelial cell-specific muscarinic receptors, such as in drug delivery systems, is desirable for safe treatment.

High AChE activity has been observed in the intestinal epithelium [89,99]. Therefore, AChE inhibitors may be effective at maintaining intestinal epithelial homeostasis by increasing ACh stability near the intestinal epithelial cells and enhancing ACh signaling. In fact, the inhibition of cholinesterase by paraoxon has been demonstrated to promote the degranulation of goblet and Paneth cells, which promotes defense against orally administered *Salmonella* [100]. The acetylcholinesterase inhibitor, pyridostigmine, was also found to attenuate the pathology of DSS-induced colitis [101].

The promotion of ACh synthesis and release also enhances ACh signaling. Brain orexins stimulate the vagal cholinergic pathway, which prevents LPS-induced colonic epithelial permeation in an mAChR-dependent manner [102]. As ACh-producing tuft cells express chemoreceptors and respond to bitter substances, ACh release may be regulated by dietary components [84]. Moreover, the short-chain fatty acid generated by intestinal bacteria, propionate, induces the secretion of non-neuronal ACh in the colon [103,104]. Therefore, prebiotics or probiotics may contribute to the upregulation of non-neuronal ACh in the colonic epithelium. Importantly, the effect of propionate is expected to be specific to intestinal epithelial cells, whereas AChE inhibitors are expected to enhance ACh in both neuronal and non-neuronal cells. Owing to its local action, ACh is thought to exert different effects depending on the localization of the released cells. However, the effects of epithelial cell-derived non-neuronal ACh on intestinal epithelial homeostasis and IBD are not fully understood. Therefore, the facilitative effect of propionate-mediated non-neuronal ACh release on intestinal epithelial homeostasis is an interesting subject for basic research and clinical applications.

## 10. Conclusions

In summary, mAChRs are involved in various functions related to intestinal epithelial homeostasis, in addition to the previously known intestinal contractions and secretions. However, the subtypes of mAChRs that function on different cell types in the intestinal epithelium and surrounding cell populations, including immune cells, are not fully understood. In addition, the cholinergic system that delivers ACh to the mAChRs in each cell is unclear. As a result, the complete mechanism of action of the mAChRs in the intestinal epithelium remains elusive. Although not discussed in detail in this review, nicotinic receptors are also important targets of ACh, and their involvement in the intestinal inflammatory response is a topic of intense research. Therefore, a comprehensive understanding of the ACh network throughout the intestinal tissue, including the relationship between muscarinic and nicotinic receptors, would highlight a new strategy for IBD treatment that contributes to intestinal epithelial homeostasis.

## Figures and Tables

**Figure 1 ijms-24-06508-f001:**
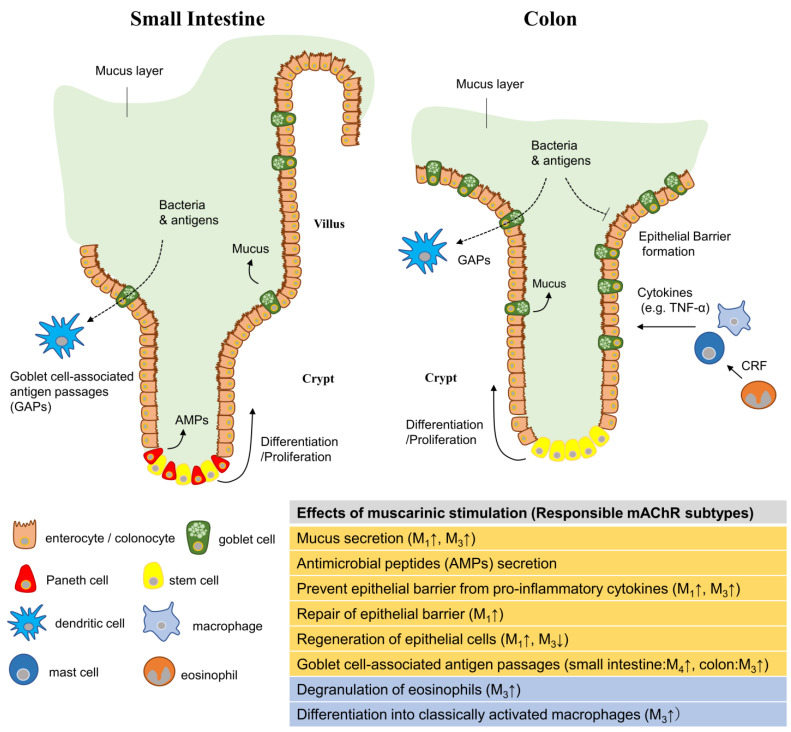
Mucosal defense system and muscarinic receptor action in intestinal epithelial tissue. The intestinal epithelium of the small intestine (left panel) and colon (right panel) are shown. Intestinal epithelial cells are composed of enterocytes/colonocytes, goblet cells, Paneth cells, and stem cells that are responsible for the formation of the epithelial barrier, secretion of mucus and antimicrobial peptides (AMPs), and replenishment of epithelial cells through differentiation and proliferation, respectively. Various immune cells reside or infiltrate the submucosal layer and are responsible for the immune response against invading bacteria. These cells express mAChRs that modulate the function of each cell. The table below summarizes the effects of mAChRs on each of these functions. The effects on epithelial cells are highlighted in orange while those on other cells are highlighted in blue. The up and down arrows marked to the right of the mAChR subtypes indicate that activating each subtype enhances or inhibits that response, respectively.

**Table 1 ijms-24-06508-t001:** mAChR subtypes in intestinal epithelial cells.

	Tissues	Region/Cell Types	Subtypes	Methods	Ref.
mouse	small intestine	villi	M_1_, M_2_, M_3_, M_4_ (M_5_-)	qPCR	[8]
		crypts	M_1_, M_3_, M_4_ (M_2_-, M_5_-)	qPCR	[8]
		epithelial cells	M_1_, M_2_, M_3_, M_4_, M_5_	IHC	[9]
			M_1_, M_3_ (M_2_-, M_4_-, M_5_-)	qPCR	[10]
			M_1_, M_2_, M_3_, M_4_, M_5_	qPCR, IHC	[11]
		goblet cells	M_3_, M_4_, (M_1_-, M_2_-, M_5_-)	qPCR, microarray	[12]
		Paneth cells	M_2_	IHC	[11]
			M_3_	IHC	[10]
		stem cells (crypt base)	M_1_ (M_2_-, M_3_-, M_4_-, M_5_-)	qPCR	[13]
			M_1_, M_3_	qPCR	[8]
			M_3_	IHC	[10]
		endocrine cells	M_3_	IHC	[10]
	colon	crypts	M_1_ (80%), M_3_ (20%)	pharmacological	[14]
			M_1_	qPCR	[15]
		goblet cell	M_3_, M_4_, (M_1_-, M_2_-, M_5_-)	qPCR, microarray	[12]
rat	small intestine	stem cell	M_3_, M_5_	IHC	[16]
human	colon	epithelial cells	M_1_	ISH	[17]
			M_3_ (M_1_-)	IHC	[18]
	cell line (colon)	HT-29/B6 cells	M_3_ (M_1_-)	pharmacological, qPCR	[19]
	cell line (colon)	T84 cells	M_3_ (65%), M_1_ (35%)	pharmacological	[20]

qPCR—quantitative PCR; IHC—immunohistochemistry; ISH—in situ hybridization. “-” indicates that they were not detected in the subtypes.

## Data Availability

No new data were created or analyzed in this study. Data sharing is not applicable to this article.

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
