# Peer review of "Role of Muscarinic Acetylcholine Receptors in Intestinal Epithelial Homeostasis: Insights for the Treatment of Inflammatory Bowel Disease"

_ijms, 2023, doi:10.3390/ijms24076508_

Round 1

Reviewer 1 Report

In this manuscript, the authors reviewed studies elucidating the distinct roles of muscarinic receptor subtypes for the maintenance of intestinal epithelial homeostasis. Furthermore, they discussed the therapeutic potential of muscarinic receptors as pharmacological targets for inflammatory bowel diseases (IBD). In general, this is a well-written comprehensive review of the field. I just have a few observations and suggestions for further improvement.

Major point:

1.       There is evidence supporting the expression of M3 muscarinic receptors in the mucosa of the human colon using immunohistochemistry (Neurogastroenterol Motil (2010) 22, 999–e263, doi: 10.1111/j.1365-2982.2009.01456.x). Thus, kindly consider adding this information to Table 1.

2.       The authors discussed the potential of M3 receptor-selective ligands for the treatment of IBD on page 8. Although activation of M3 receptors may be helpful for treating IBD by suppressing inflammatory cytokines, they may also have adverse effects on the disease prognosis by impairing the epithelial-barrier function. Furthermore, it is well known that muscarinic contractions in the gastrointestinal tract are mediated by the combined activation of M2 and M3 receptors, with a greater contribution from the M3 receptors. Thus, gastrointestinal symptoms, such as diarrhea and constipation are highly possible symptoms when selective M3 receptor ligands are for IBD treatment. Thus, special drug delivery systems or other pharmacological approaches may be required to address this problem. I suggest that the authors must discuss this aspect in the manuscript.

3.       Figure 1 appeared to be in low resolution, and the letters in the figure are small and illegible. Please revise the figure and its components.

Minor point:

1.       There seems to be a double space between “activation” and “of” in line 24.

2.       The punctuation mark (period) in the last sentence of the Abstract is doubled.

3.       Please delete the space between “M3” and “KO” in lines 194 and 218.

4.       Define the term “AMPs” in line 316.

5.       Please delete “1.” in line 661.

Author Response

Reviewer1:

 We thank Reviewer 1 for the positive comments. We revised the manuscript as follows.

Major point:

  1. There is evidence supporting the expression of M3 muscarinic receptors in the mucosa of the human colon using immunohistochemistry (Neurogastroenterol Motil (2010) 22, 999–e263, doi: 10.1111/j.1365-2982.2009.01456.x). Thus, kindly consider adding this information to Table 1.

Thank you for your suggestion. Evidence of muscarinic receptor expression in human colon samples is very important from a clinical perspective. We have added the proposed paper to Table 1.

  1. The authors discussed the potential of M3 receptor-selective ligands for the treatment of IBD on page 8. Although activation of M3 receptors may be helpful for treating IBD by suppressing inflammatory cytokines, they may also have adverse effects on the disease prognosis by impairing the epithelial-barrier function. Furthermore, it is well known that muscarinic contractions in the gastrointestinal tract are mediated by the combined activation of M2 and M3 receptors, with a greater contribution from the M3 receptors. Thus, gastrointestinal symptoms, such as diarrhea and constipation are highly possible symptoms when selective M3 receptor ligands are for IBD treatment. Thus, special drug delivery systems or other pharmacological approaches may be required to address this problem. I suggest that the authors must discuss this aspect in the manuscript.

According to the suggestion, we added following sentence in lines 398-402.

“Gastrointestinal disorders, such as diarrhea are major adverse effects of muscarinic stim-ulation. This effect is mainly due to the M3 subtype in smooth muscle, despite reports of the involvement of the M1 and M2 subtypes [97, 98]. Therefore, the stimulation of epithelial cell-specific muscarinic receptors, such as in drug delivery systems, is desirable for safe treatment.”

  1. Figure 1 appeared to be in low resolution, and the letters in the figure are small and illegible. Please revise the figure and its components.

We revised Figure 1 and made the letters larger.

Minor point:

  1. There seems to be a double space between “activation” and “of” in line 24.

The indicated point was corrected in revised version.

  1. The punctuation mark (period) in the last sentence of the Abstract is doubled.

The indicated point was corrected in revised version.

  1. Please delete the space between “M3” and “KO” in lines 194 and 218.

The space was added in revised version.

  1. Define the term “AMPs” in line 316.

The definition of AMPs had been included in the text that was lost during the submission process. The revised manuscript includes this missing part.

  1. Please delete “1.” in line 661.

We deleted “1.” at the end of the manuscript.

Reviewer 2 Report

In this paper, Uwada et al. review advances in research exploring the role of muscarinic acetylcholine receptors in maintaining intestinal epithelial homeostasis, the involvement of non-neuronal acetylcholine release, and the potential for using this information to develop novel therapeutic strategies to treat IBD. Overall, this is an important and timely topic for review. However, several issues, primarily the organizational structure of the paper and English usage need to be addressed before the manuscript is ready for publication.

Major Comments

1.    Abstract: The authors might acknowledge early on that what is referred to generally as IBD is comprised of important subsets, most notably Crohn’s disease and ulcerative colitis. It is incorrect to state that there is ‘truly no effective treatment for the disease.’  Indeed, the past 10-15 years has witnessed the development and emergence of several new effective treatments.

2.    Accepted nomenclature is for subscript numerals for muscarinic receptor subtypes (e.g., M1R).

3.    The writing too often lacks correct English usage and is difficult to follow. Examples of incorrect English usage: line 37 (‘other antigens, leading to trigger inflammatory responses’) and line 39 (‘impaired intestinal barrier function exacerbated inflammation’) and line 173 (‘kinase 1/2 (ERK1/2) activation and following focal adhesion kinase (FAK) activation’) and many others. Editing with someone better versed in English usage is likely to be beneficial.

4.    The discussion in lines 50-55 appears somewhat confused regarding the distinction between nicotinic and muscarinic receptors.

5.    Although the authors persistently refer to McN-A-343 as ‘M1R-selective’ they should acknowledge that some studies dispute this – e.g., see Eglen et al. Br. J. Pharmac. 1987;90:693-700.

6.    Figure 1 and related discussion: The relevance of intestinal barrier function regarding IBD is generally focused on the colon, not the small intestine. Much, if not all, of the experimental data regarding this are derived from studies of the colon, not small intestine. Hence, it appears to this reviewer that the figure should focus on the colon, not small intestine. This requires revision to remove Paneth cells which are not expressed in the normal colon and other features that may not be relevant to human IBD and barrier function in the colon.

7.    The final sentence of the Conclusions section is highly dramatic and speculative. There is absolutely no guarantee that such an understanding will deliver therapeutic agents that will be more effective than currently available therapy. This needs to be toned down.  

 Minor Comments

1.    Line 26: Double periods at the end of the sentence.

2.    Line 153: It is not clear what ‘In contrast’ refers to.

Author Response

Reviewer2

We thank Reviewer1 for the valuable comments. We revised the manuscript as follows.

Major Comments

  1. Abstract: The authors might acknowledge early on that what is referred to generally as IBD is comprised of important subsets, most notably Crohn’s disease and ulcerative colitis. It is incorrect to state that there is ‘truly no effective treatment for the disease.’ Indeed, the past 10-15 years has witnessed the development and emergence of several new effective treatments.

In the abstract, we have added a statement that IBD includes Crohn's disease and ulcerative colitis. The statement "truly no effective treatment for the disease." was intended to emphasize that although several treatments have been developed to lead into remission, there is still no treatment that provides a complete cure, and that it remains important to understand the cause of IBD and to develop a new treatment. The statement was revised in the manuscript.

  1. Accepted nomenclature is for subscript numerals for muscarinic receptor subtypes (e.g., M1R).

The numerals representing each subtype have been changed to subscripts.

  1. The writing too often lacks correct English usage and is difficult to follow. Examples of incorrect English usage: line 37 (‘other antigens, leading to trigger inflammatory responses’) and line 39 (‘impaired intestinal barrier function exacerbated inflammation’) and line 173 (‘kinase 1/2 (ERK1/2) activation and following focal adhesion kinase (FAK) activation’) and many others. Editing with someone better versed in English usage is likely to be beneficial.

We apologize for the confusing text. The revised manuscript was edited by the native English speaker.

  1. The discussion in lines 50-55 appears somewhat confused regarding the distinction between nicotinic and muscarinic receptors.

We have rewritten the sections you pointed out to avoid confusion and to make them clearer (lines 50-59).

  1. Although the authors persistently refer to McN-A-343 as ‘M1R-selective’ they should acknowledge that some studies dispute this – e.g., see Eglen et al. Br. J. Pharmac. 1987;90:693-700.

As you have pointed out, the action of McN-A-343 is not a pure M1 selective agonist and seems to exhibit complex behavior. In the revised manuscript, we have added the following statement and added the relevant references (lines 206-210).

“McN-A-343 is widely used as an M1 selective agonist owing to its relatively high efficacy against the M1 subtype; however, it also acts on other mAChR subtypes [63].”

ref. 63. Mitchelson, F. J., The pharmacology of McN-A-343. Pharmacol Ther 2012, 135, 216-45.

  1. Figure 1 and related discussion: The relevance of intestinal barrier function regarding IBD is generally focused on the colon, not the small intestine. Much, if not all, of the experimental data regarding this are derived from studies of the colon, not small intestine. Hence, it appears to this reviewer that the figure should focus on the colon, not small intestine. This requires revision to remove Paneth cells which are not expressed in the normal colon and other features that may not be relevant to human IBD and barrier function in the colon.

We apologize that the text describing the important role of mAChR in Paneth cells and goblet cells was lost in the process of transferring the original text to the IJMS submission form. In the revised manuscript, the lost part has been restated as originally described. The function of Paneth cells and goblet cells is critical for intestinal homeostasis and IBD symptoms. Thus, we had included these cells in the Figure 1. For simplicity, the illustrated intestinal epithelium had combined characteristics of small intestine and colon. However, this simplification may be misleading. In the revised version, we have divided the illustration into small intestine and colon.

  1. The final sentence of the Conclusions section is highly dramatic and speculative. There is absolutely no guarantee that such an understanding will deliver therapeutic agents that will be more effective than currently available therapy. This needs to be toned down.

According to the suggestion, we rewrote the last sentence in the revised version (lines 446-448).

 Minor Comments

  1. Line 26: Double periods at the end of the sentence.

The indicated point was corrected in revised version.

  1. Line 153: It is not clear what ‘In contrast’ refers to.

‘In contrast’ in Line 153 was deleted (line 214).

Round 2

Reviewer 2 Report

The authors have been responsive to the comments in the initial review.

Minor comment: The word 'crypt' is misspelled twice in Fig. 1.

Author Response

We thank reviewer2 for the positive comment. We corrected the manuscript as follows.

The word 'crypt' is misspelled twice in Fig. 1.

  →Thank you for a valuable suggestion. We corrected this point.